# The Allelopathic Activity of Extracts and Isolated from *Spirulina platensis*

**DOI:** 10.3390/molecules27123852

**Published:** 2022-06-15

**Authors:** Patchanee Charoenying, Chamroon Laosinwattana, Nawasit Chotsaeng

**Affiliations:** 1Department of Chemistry, School of Science, King Mongkut’s Institute of Technology Ladkrabang, Bangkok 10520, Thailand; nawasit.ch@kmitl.ac.th; 2Department of Plant Production Technology, School of Agricultural Technology, King Mongkut’s Institute of Technology Ladkrabang, Bangkok 10520, Thailand; klchamro@kmitl.ac.th; 3Integrated Applied Chemistry Research Unit, School of Science, King Mongkut’s Institute of Technology Ladkrabang, Bangkok 10520, Thailand

**Keywords:** allelopathic effects, *Spirulina platensis*, cyanobacteria, allelochemical, Chinese amaranth, barnyardgrass, weeds

## Abstract

We determined the allelopathic effects of crude organic (hexane, ethyl acetate, and methanol) extracts of the cyanobacterial *Spirulina platensis* on barnyardgrass (*Echinochloa crus-galli* (L.) Beauv.) and Chinese amaranth (*Amaranthus tricolor* L.). The crude ethyl acetate extract showed the highest inhibitory activity and was subsequently fractionated by column chromatography into 23 fractions based on thin-layer chromatography band pattern similarities. Four concentrations (2000, 1000, 500, and 250 ppm) of each fraction were tested for their allelopathic activity. Fractions E6 and E13 exhibited the most significant inhibitory effects against Chinese amaranth. The constituents of the highly active E6F3-E6F5 fractions determined by GC-MS, chromatography, and spectroscopy included the fatty acids, γ-linolenic acid **15**, oleic acid **12**, and predominantly palmitic acid **7**; minor constituents included 2-ethyl-3-methylmaleimide **9** and C_11_ norisoprenoids (dihydroactinidiolide **10** and 4-oxo-β-ionone **13**). Isolation of E13 fraction by column chromatography revealed four C_13_ norisoprenoids: 3-hydroxy-β-ionone **17**, 3-hydroxy-5α,6α-epoxy-β-ionone **18**, 3-hydroxy-5β,6β-epoxy-β-ionone **19**, and loliolide **20**. Their structures were elucidated by NMR spectroscopy. All six isolated norisoprenoids inhibited seed germination and seedling growth of Chinese amaranth at concentrations of 250–1000 ppm. Allelochemicals from *S. platensis* could be utilized in the development of novel bioactive herbicides.

## 1. Introduction

One major obstacle to improving the world’s agricultural efficiency is weed control. Weeds are simply those plants growing in an undesired location, and they complete with crops for resources, decrease crop yield, and contaminate cropland with their seeds, thereby perpetuating the problem into subsequent growing seasons [1,2]. Preventive, chemical, biological, mechanical, and cultural methods can be used to control weeds. The appropriate approach depends on many factors, including the species of weed being managed, levels of weed infestation, and available resources, such as time, personnel, equipment and funding. The main weed-management method is chemical control by using herbicides [3]. However, the widespread use of synthetic herbicides has resulted in herbicide-resistant weeds, and public concerns over the impact of synthetic herbicides on human health and the environments are increasing. These concerns are shifting attention to alternative weed-control technologies based on natural products such as those from other plants that can reduce weed growth via allelopathy [4,5]. 

Most commonly defined as the direct or indirect effect of one plant on the germination, growth, or development of another plant through the production of chemicals released into the environment, allelopathy has been implicated in the patterning of vegetation and weed growth in agricultural systems and inhibition of crop development [6,7]. Chemicals released from plants that exert allelopathic influences allelochemicals or allelochemics are usually classified as secondary metabolites produced as by-products of a plant’s primary metabolic pathway. Their functions in the plant are largely undiscovered, although some are known to demonstrate structural functions or play a general defensive role against herbivores and plant pathogens [8,9,10]. Allelopathy in agriculture is currently being explored to determine the potential of allelochemicals to act as natural pesticides, providing environmentally friendly weed control that is renewable and easily degradable [11,12,13,14,15,16,17,18].

Allelopathic phenomena are routinely recognized in terrestrial plants, but very little is known about allelopathy in marine plants [19]. However, it has been observed that microorganisms, such as algae and some cyanobacterial allelochemicals, have been isolated and characterized [20,21,22,23,24,25,26,27]. Previous studies have reported that cyanobacterial allelopathy played a role in phytoplankton succession by affecting some species more than others [6]. Furthermore, some cyanobacteria have demonstrated allelopathic properties on plants by affecting their germination and growth. One example, *Nostoc* 31, exhibited allelopathic activity against cyanobacteria *Anabaena* 7120 [28]. Moreover, Martin and co-workers found that the blue-green algae *Gomphosphaeria aponina* showed allelopathic potential against the red tide algae *Gymnodinium breve* but not vice versa [29].

The blue-green algae *Spirulina platensis*, a photosynthetic cyanobacterium containing many nutrients and exhibiting numerous biological activities, has been the subject of much research, not only for its human nutritional characteristics but also for the development of potential pharmaceuticals [30,31,32]. We previously reported the allelopathic effects of its aqueous extract, C-phycocyanin, and crude organic, (hexane, ethyl acetate, and methanol) extracts on select plant species. The aqueous extract demonstrated potent inhibitory activity and C-phycocyanin inhibited Chinese amaranth and Chinese mustard. The crude ethyl acetate extract showed the highest inhibitory activity against Chinese amaranth, Chinese mustard, barnyardgrass, and rice [33,34]. In the current study, we fractionated, isolated, and identified these compounds in the crude ethyl acetate extract to better understand this inhibitory activity and characterize the active compounds. The allelochemical compounds extracted were investigated for their allelopathic effects on Chinese amaranth and barnyardgrass. 

## 2. Results

The allelopathic screening bioassay of fractions E1–E23 (Data shown in Appendix A from the ethyl acetate extract of *S. platensis* revealed that E6 and E13 fractions demonstrated the most significant inhibitory activity [33]. In the investigation, Chinese amaranth was selected as a representative dicotyledon and barnyardgrass as a representative monocotyledon. At the highest applied concentration (2000 ppm), the E6 fraction inhibited seed germination, shoot height, and root growth (length) of Chinese amaranth by 84.85%, 90.28%, and 89.95%, respectively. The E13 fraction completely inhibited Chinese amaranth seed germination and seedling growth. In barnyardgrass, the E6 fraction inhibited seed germination, shoot, and root growth by 92.11%, 49.47%, and 72.12%, respectively, whereas the E13 fraction only minimally inhibited barnyardgrass.

### 2.1. Allelopathic Effects of the Seven E6 Subfractions

Seven subfractions were obtained from the E6 fraction, and their allelopathic subfraction activity against Chinese amaranth and barnyardgrass was measured at 125–1000 ppm subfraction concentrations. Figure 1 and Figure 2 show that subfractions E6F3-E6F5 demonstrated substantial inhibition against both tested plants. At a concentration of 500 ppm, subfraction E6F4 completely inhibited seed germination of Chinese amaranth and at the highest concentration (1000 ppm), it inhibited seed germination, shoot growth, and root growth of barnyardgrass, by 94.59%, 95.69%, and 100%, respectively. 

### 2.2. Allelopathic Effects of Six E13 Subfractions

As shown in Figure 3 and Figure 4, subfractions E13F2-E13F4 demonstrated significant inhibition of seed germination and seedling growth in Chinese amaranth at concentrations of 500–1000 ppm. The most inhibitory subfraction was E13F3, which completely inhibited Chinese amaranth seed germination at a concentration of 500 ppm. In contrast, subfractions E13F1 and E13F5-E13F6 moderately stimulated the growth root of Chinese amaranth. Furthermore, although all subfractions slightly inhibited seed germination and seedling growth of barnyardgrass, subfractions E13F1 and E13F5 stimulated barnyardgrass root growth.

### 2.3. Isolation and Identification of Compounds

The active subfractions E6F3-E6F5 and E13F1-E13F4 were isolated and pure compounds were characterized by chromatographic and spectroscopic techniques. These major compounds were divided into two groups, fatty acids, and norisoprenoids. These results represent the first report of these norisoprenoids from *S. platensis*. The crude extract demonstrating the highest inhibitory activity (ethyl acetate) was subjected to short column chromatography (CC) on silica gel, using a gradient solvent system of *n*-hexane, *n-*hexane-EtOAc, EtOAc-MeOH, and MeOH in increasing polar solvent proportion to produce 23 fractions (E1-E23). The fractions were tested for allelopathic activity on Chinese amaranth and barnyardgrass at concentrations of 250–2000 ppm. The most potent active fractions, E6 and E13, were further separated by CC and flash column chromatography (FCC). E6 (1.2 g) was fractionated by CC on silica gel by using a gradient solvent system of *n*-hexane-EtOAc (95:5) and EtOAc in increasing proportions of the polar solvent to yield seven subfractions (E6F1-E6F7) based on their TLC behavior. The subfractions were tested for allelopathic activity on the two test plant species at concentrations of 125–1000 ppm. E6F3 (255.2 mg) eluted with *n*-hexane-EtOAc 96:4), E6F4 (105.2 mg, eluted with *n*-hexane-EtOAc 96:4–95:5) and E6F5 (186.5 mg, *n*-hexane-EtOAc 95:5) showed the highest inhibitory activity. 

Fifty milligrams each of subfractions E6F3, E6F4, and E6F5 were analyzed and the active compounds were identified by GC-MS as methyl ester derivatives (compounds **1**–**16**, Table 1, Figure 5). E6F3-E6F5 contained fatty acid methyl esters, (identified by GC-MS using heptadecanoic acid methyl ester as an internal standard). The GC-MS results of active subfractions revealed that subfraction E6F3 was composed of palmitic acid **7** (58.71%), palmitoleic acid **8** (12.82%), oleic acid **12** (11.76%), tetradecanoic acid **3** (3.97%), pentadecanoic acid **4** (2.16%), and the minor constituents, dodecanoic acid **1**, 9-oxonanoic acid **5**, azelaic acid **6**, dihydroactinidiolide **10**, stearic acid **11**, 4-oxo-β-ionone **13**, and linoleic acid **14**. The most active subfraction, E6F4, consisted of linoleic acid **14** (72.19%), dihydroactinidiolide **10** (8.32%), palmitoleic acid **8** (4.55%), γ-linolenic acid **15** (3.02%), 2-ethyl-3-methylmaleimide **9** (2.90%), 5,8,11-eicosatrienoic acid **16** (2.53%), palmitic acid **7** (2.29%), 4-oxo-β-ionone **13** (2.15%), and the minor constituents, 9-oxonanoic acid **5**, steric acid **11**, and oleic acid **12**. The active subfraction E6F5 consisted of palmitic acid **7** (31.41%), linoleic acid **14** (31.24%), γ-linolenic acid **15** (23.46%), 5,8,11-eicosatrienoic acid **16** (2.80%), oleic acid **12** (3.16%), palmitoleic acid **7** (2.71%), 4-oxo-β-ionone **13** (1.91%), and the minor constituents, neophytadiene, tetradecanoic acid **3**, and stearic acid **11**. E6F3, E6F4, and E6F5 were eluted using FCC to yield dihydroactinidiolide **10** (9.6 mg) and 4-oxo-β-ionone **(13)** (7.2 mg). 

Fraction E13 (650.0 mg) was re-eluted by CC on silica gel by using *n-*hexane/EtOAc (95:5) and EtOAc to increase the proportions of the polar solvent. The six resulting subfractions were screened for allelopathic activity at concentrations of 125–1000 ppm. E13F2 (56.4 mg, eluted with *n*-hexane/EtOAc 96:4), E13F3 (112.5 mg, eluted with *n*-hexane/EtOAc 96:4–95:5), and E13F4 (88.3 mg, eluted with *n*-hexane/EtOAc 95:5) exhibited high inhibitory activity. Fifty milligrams of subfraction E13F2, E13F3, and E13F4 were combined and treated by FCC by using silica gel 60. The mixture was eluted by using an *n*-hexane/EtOAc gradient to obtain four known compounds shown in Figure 6, 3-hydroxy-β-ionone **17** (27.0 mg), 3-hydroxy-5α,6α-epoxy-β-ionone **18** (7.5 mg), 3-hydroxy-5β,6β-epoxy-β-ionone **19** (6.9 mg), and loliolide **20** (35.0 mg). By comparison of their spectral analyses with those reported in the literature, these compounds were identified as dihydroactinidiolide **10**, 4-Oxo-β-ionone **13**, 3-Hydroxy-β-ionone **17**, 3-Hydroxy-5α, 6α-epoxy-β-ionone **18**, 3-Hydroxy-5β, 6β-epoxy-β-ionone **19**, and Loliolide **20**. The isolated compounds were identified by ^1^H NMR and ^13^C NMR as shown below.

*Dihydroactinidiolide***10**, colorless oil, 9.6 mg (0.027% from crude ethyl acetate extract); ^1^H NMR (CDCl_3_) 1.22 (3H, *s*, H-9), 1.27 (3H, *s*, H-10), 1.22–1.34 (1H, *m*, H-2α), 1.46 (1H, *m*, H-4α), 1.55 (3H, *s*, H-11), 1.62–1.72 (1H, *m*, H-2β), 1.72–1.77 (2H, *m*, H-3), 2.24 (1H, *dq*, *J* = 12.4, 2.0 Hz, H-4β), 5.64 (1H, *s*, H-7). ^13^C NMR (CDCl_3_), 19.65 (C-3), 24.18 and 29.83 (C-9 and C-10), 24.35 (C-11), 36.50 (C-1), 40.09 (C-4), 41.66 (C-2), 87.26 (C-5), 112.38 (C-7), 171.99 (C-6), 182.51 (C-8) [35,36,37]. 

*4-Oxo-**β-ionone***13**, waxy solid, 7.2 mg (0.018% from crude ethyl acetate extract); ^1^H NMR (CDCl_3_) 1.19 (6H, *s*, H-11 and H-12), 1.80 (3H, *s*, H-13), 1.89 (2H, *t*, *J* = 6.8 Hz, H-2), 2.35 (3H, *s*, H-10), 2.54 (2H, *t*, *J* = 6.8 Hz, H-3), 6.19 (1H, *d*, *J* = 16.5 Hz, H-8), 7.24 (1H, *d*, *J* = 16.5 Hz, H-7). ^13^C NMR (CDCl_3_), 13.43 (C-13), 27.35 (2C, C-11 and C-12), 27.94 (C-10), 34.21 (C-3), 35.60 (C-1), 37.41 (C-2), 131.14 (C-5), 133.57 (C-8), 140.31 (C-7), 157.16 (C-6), 198.00 (C-9), 198.50 (C-4) [38,39].

*3-Hydroxy-**β-ionone***17**, pale yellow oil, 27 mg (0.076% from crude ethyl acetate); ^1^H NMR (CDCl_3_) 1.11 (3H, *s*, H-12), 1.12 (3H, *s*, H-11), 1.50 (1H, *t*, *J* = 11.9 Hz, H-2α), 1.65 (1H, *br*-OH), 1.78 (3H, *s*, H-13), 1.78–1.82 (1H, *m*, H-2β), 2.09 (1H, *dd*, *J* = 17.6, 9.40 Hz, H-4α), 2.30 (3H, *s*, H-10), 2.44 (1H, *dd*, *J* = 17.4, 5.5 Hz, H-4β), 3.96–4.06 (1H, *m*, H-3α), 6.12 (1H, *d*, *J* = 16.4 Hz, H-8), 7.21 (1H, *d*, *J* = 16.3 Hz, H-7). ^13^C NMR (CDCl_3_) 21.59 (C-13), 27.32 (C-10), 28.58 and 30.09 (C-11 and C-12), 36.92 (C-1), 42.78 (C-4), 48.44 (C-2), 64.54 (C-3), 132.26 (C-5), 132.39 (C-8), 135.66 (C-6), 142.33 (C-7), 198.52 (C-9) [40,41].

*3-Hydroxy-5**α, 6**α-epoxy-**β-ionone***18**, colorless oil, 7.5 mg (0.021% from crude ethyl acetate); ^1^H NMR (CDCl_3_) 0.98 (3H, *s*, H-12), 1.19 (6H, *s*, H-11 and H-13), 1.24 (1H, *dd*, *J* = 13.1, 2.6 Hz, H-2α), 1.60–1.70 (1H, *br*-OH), 1.62–1.67 (1H, *m*, H-2β), 1.65–1.69 (1H, *m*, H-4α), 2.29 (3H, *s*, H-10), 2.40 (1H, *ddd*, *J* = 14.3, 5.0, 1.5 Hz, H-4β), 3.86–3.95 (1H, *m*, H-3α), 6.29 (1H, *d*, *J* = 15.5 Hz, H-8), 7.03 (1H, *d*, *J* = 15.5 Hz, H-7). ^13^C NMR (CDCl_3_) 19.87 (C-13), 25.00 (C-12), 28.29 (C-10), 29.35 (C-11), 35.12 (C-1), 40.60 (C-4), 46.68 (C-2), 64.02 (C-3), 67.24 (C-5), 69.49 (C-6), 132.63 (C-8), 142.35 (C-7), 197.38 (C-9) [42,43].

*3-Hydroxy-5**β, 6**β-epoxy-**β-ionone***19**, colorless oil, 6.9 mg (0.020% from crude ethyl acetate); ^1^H NMR (CDCl_3_) 1.00 (3H, *s*, H-11), 1.19 (3H, *s*, H-13), 1.21 (3H, *s*, H-12), 1.37 (1H, *dd*, *J* = 12.9, 3.5 Hz, H-2β), 1.56–1.64 (1H, *m*, H-2α), 1.56–1.64 (1H, *br*-OH), 1.89 (1H, *dd*, *J* = 14.8, 8.5 Hz, H-4α), 2.23 (1H, *dd*, *J* = 14.6, 6.4 Hz, H-4β), 2.28 (3H, *s*, H-10), 3.85–3.95 (1H, *m*, H-3α), 6.31 (1H, *d*, *J* = 15.6 Hz, H-8), 6.98 (1H, *d*, *J* = 15.6 Hz, H-7). ^13^C NMR (CDCl_3_) 20.98 (C-13), 25.78 (C-12), 26.84 (C-11), 28.49 (C-10), 34.86 (C-1), 38.92 (C-4), 43.51 (C-2), 63.73 (C-3), 66.00 (C-5), 71.00 (C-6), 133.13 (C-8), 141.02 (C-7), 197.21 (C-9) [44,45].

*Loliolide***20**, needle solid (crystallized from hexane/EtOAc) 35 mg (0.099% from crude ethyl acetate); ^1^H NMR (CDCl_3_) 1.27 (3H, *s*, H-10), 1.47 (3H, *s*, H-9), 1.52 (1H, *dd*, *J* = 14.6, 3.6 Hz, H-2α), 1.79 (3H, *s*, H-11), 1.79 (1H, *dd*, *J* = 13.9, 3.9 Hz, H-4α), 1.95 (1H, *br*-OH), 2.00 (1H, *dt*, *J* = 14.4, 2.5 Hz, H-2β), 2.48 (1H, *dt*, *J* = 13.9, 2.3 Hz, H-4β), 4.31–4.34 (1H, *m*, H-3α), 5.69 (1H, *s*, H-7). ^13^C NMR (CDCl_3_) 26.50 (C-9), 27.03 (C-11), 30.67 (C-10), 35.98 (C-1), 45.66 (C-4), 47.33 (C-2), 66.67 (C-3), 86.92 (C-5), 112.79 (C-7), 172.08 (C-6), 182.79 (C-8) [46,47,48].

### 2.4. Allelopathic Effects of Pure Compounds

Compounds from the subfractions demonstrating allelopathic activity were categorized as fatty acids (Table 1) or norisoprenoids. Commercial fatty acids **7**, **11**, **12**, **14**, and **15** were selected to represent primary natural fatty acids from *S. platensis*. Isolated norisoprenoids **10**, **13**, and **17**–**20** were tested for allelopathic activity at concentrations of 62.5–1000 ppm (Figure 7). Linoleic acid **14** and γ-linolenic acid **15** at concentrations of 500–1000 ppm were highly inhibitory to seed germination of Chinese amaranth. Both fatty acids completely inhibited seed germination at a concentration of 1000 ppm shoot growth at 500–1000 ppm and root growth at 250–1000 ppm. In contrast, palmitic acid **7**, steric acid **11**, and oleic acid **12** stimulated Chinese amaranth shoot and root growth.

Linoleic acid **14** and γ-linolenic acid **15** exhibited significant barnyardgrass seed germination at 500–1000 ppm (Figure 8). Oleic acid **12** at a concentration of 1000 ppm slightly inhibited shoot growth, and linoleic acid **14** and γ-linolenic acid **15** at 500–1000 ppm moderately reduced barnyardgrass shoot and root growth. Palmitic acid **7**, steric acid **11**, and oleic acid **12** were weak promotors of seedling growth.

### 2.5. Allelopathic Effects of Norisoprenoids

Six C_11_ and C_13_ norisoprenoids, exhibited different allelopathic effects on the two test plant species. The results (Figure 9) show that dihydroactinidiolide **10**, 3-hydroxy-β-ionone **17**, and 3-hydroxy-5α, 6α-epoxy-β-ionone **18** at concentrations of 250–1000 ppm, and 4-oxo-β-ionone **13**, 3-hydroxy-5β, 6β-epoxy-β-ionone **19** and loliolide **20** at concentrations of 500–1000 ppm were highly inhibitory of Chinese amaranth seed germination. Compounds **10**, **18**, and **19** completely inhibited at a concentration of 500 ppm. Compounds **10**, **17**, **18**, and **19** at 125–1000 ppm and compounds **13** and **20** at 500–1000 ppm substantially inhibited Chinese amaranth shoot growth. Similarly, compounds **10** and **13** at 125–1000 ppm, compounds **17**, **18**, and **19** at 250–1000 ppm, and compounds **13** at concentrations of 500–1000 ppm inhibited the root growth of Chinese amaranth. Furthermore, it was noted that compound **18** had a higher allelopathic effect on Chinese amaranth than compound **19**.

The allelopathic effects on barnyardgrass were determined for compounds, **10**, **13**, **17**, and **20**. Compounds **18** and **19** were not assessed because the compounds constituted relatively small proportions of the total fractional composition. Compound **10** at 250–1000 ppm, compound **13** at 500–1000 ppm, and compound **17** at 1000 ppm inhibited seed germination of barnyardgrass (Figure 10). Notably, compound **10** at the highest concentration exhibited complete inhibition of barnyardgrass seed germination; however, the other concentrations and loliolide **20** did not. Shoot growth was inhibited by compound **10**, compound **17** at 125–1000 ppm, and compounds **13** and **20** at 500–1000 ppm inhibited shoot growth of barnyardgrass; other concentrations had no effect. Compounds **10**, **13**, and **17** at 250–1000 ppm and compound **20** at a concentration of 1000 ppm inhibited barnyardgrass root growth.

## 3. Discussion

Fatty acids have long been explored for their allelopathic effects. In one of the earliest studies, in 1985 Browers and coworkers [49] isolated the allelochemical laetisaric acid {(Z, Z)-9,12-8-hydroxyoctadecadienoic acid)} from the fungus, *Laetisaria arvalis*. At 22 µg/mL, this acid inhibited the growth of the plant pathogen *Pythium ultimum* by 50%. Kakisawa and coworkers [19] subsequently isolated an allelopathic substance from the brown algae *Chadosiphon okamuranus*. Identified as 6*Z*,9*Z*,12*Z*,15*Z*-octadecatetraenoic acid, the compound inhibited the growth of the red algae *Porphyra yezoensis*, and the microalgae *Heterosigma akashiwo*. Phytotoxic compounds have also been isolated from the aqueous and leachates of cattails (*Typha domingensis*). These compounds were identified as essential fatty acids (linoleic acid 14 and α–linolenic acid) and phenolic compounds. The two fatty acids were the predominant components in the active fractions, comprising more than 80% of the total isolated material and these fractions inhibited the germination of lettuce seeds by 80–90% [50]. In an examination of the allelopathic potential of *Eichhornia crassipes* root extract on the cyanobacterial, *Chlorella* sp., and *Scendesmus obliquus*, Jin and coworkers [51] identified the active fraction as a saturated fatty acid and pelargonic acid (C_9:0_). This fraction reduced chlorophyll-a up to 95.3% in the target species. Rice husks were also observed to exert allelopathic effects, specifically on barnyardgrass, the most potent inhibitory effect was caused by a fraction composed of 9-octadecenoic acid, 7-octadecenoic acid, 5,8,11-heptadecatriecenoic acid, and androstan-17-one. This fraction completely inhibited seed germination of barnyardgrass at a concentration of 200 ppm and demonstrated a minimum inhibition concentration (MIC) of 50 ppm [52]. Alamsjah and coworkers [53] studied the algicidal activity of the green algae, *Ulva fascica*, and isolated three active compounds, the polyunsaturated fatty acids, hexadec-4,7,10,13-tetraenoic acid, octadeca-6,9,12,15-tetraenoic acid, and α-linolenic acid. These polyunsaturated fatty acids showed potent algicidal activity against red-tide phytoplankton *Heterosigma akshiwo* (LC_50_ 1.35 µg/mL, 0.83 µg/mL, and 1.13 µg/mL). Phytotoxic substances have also been identified from the early growth of barnyardgrass root exudates by Xuan and coworkers [54]. They found that the active fractions consisted of two phenolic derivatives, four derivatives of phthalic acid, a benzoic acid derivative, derivatives of decane, the derivative of acenaphthene, two lactones, and three long-chain fatty acids, including decanoic acid (C_10:0_), myristic acid and stearic acid **11**. At a concentration of 100 ppm, these three fatty acids showed approximately 4–20% allelopathic activity against barnyardgrass seed germination and seedling growth. In 1984, Spruell [55] studied the activity of three C_18_ fatty acids against several algae and zooplankton species. He found that the three acids, γ-linolenic acid **15**, linoleic acid **14**, and oleic acid **12**, reduced the growth of *Haematoccus lacustris*, *Synechococcus leopolienesis*, and *Botrydiopsis alpine* by 50% at concentrations below 7 ppm. 

The authors found that unsaturated fatty acids extracted from *S. platensis* demonstrated a higher inhibitory effect on Chinese amaranth and barnyardgrass than saturated fatty acids. These results are consistent with previous research that reported the allelopathic activities of fatty acids depend on (i) the length of the carbon chain, (ii) the number of unsaturated linkages, and (iii) the positions of any double bonds. For examples, Kakisawa and coworkers [19] studied the allelopathic activity of octadeca-6,9,12,15-tetraenoic acid (ODTA), arachidonic acid, 5,8,11,14-ecosapentaenoic acid (EPA), γ-linolenic acid 15, linoleic acid 14, oleic acid 12, ODTA methyl ester, methyl linolenate, ODTA sodium salt, sodium linolenate, trilinolenine, sodium laurylbenzenesulfonate, and Tween 80 against the potentially toxic algae *Heterosigma akashiwo*. They found that arachidonic acid, and EPA exhibited similar activity to ODTA. The sodium salt of ODTA was as effective an allelopathic compound as free ODTA. More saturated fatty acids exhibited less activity. The ester of unsaturated fatty acids also exhibited more worthless activities than the corresponding free acids. Sodium lauryl benzenesulfonate and Tween 80 showed inhibitory activity to a lesser degree. Aliotta and coworkers [56] isolated allelochemical compounds from *Typha latifolia* L. that consisted of three steroids and three fatty acids (α-linolenic acid, linoleic acid 14, and an unidentified C_18:2_ fatty acid). They found that the fatty acid most effective in inhibiting algae growth was α-linolenic acid. Murakami and coworkers [57] studied the allelopathic activity of the algae *Phormidium tenue* in its own culture. The active fraction (MIC 0.5 ppm) was a mixture of methyl myristate, methyl palmitate, methyl palmitoleate, methyl oleate, methyl *cis*-vaccenate, methyl linoleate, and methyl linolenate in a ratio of 5:4:5:4:1:47:36. They also examined the allelopathic effect of authentic fatty acids and found that unsaturated fatty acids such as palmitoleic acid **8**, oleic acid **12**, and *cis*-vaccenic acid were inhibitory at 2.5, 1.0, and 5.0 ppm, respectively. The more unsaturated fatty acids, linoleic acid 14 and γ-linolenic acid **15**, demonstrated the highest inhibitory activity at 0.5 ppm, while saturated fatty acids were inactive even at 100 ppm. Suzuki and coworkers [58] studied the growth-inhibitory activity of two isolated fatty acids, palmitic acid 8 and (5*Z*, 8*Z*, 11*Z*, 14*Z*, 17*Z*)-eicosapentaenoic acid (from the red algae, *Neodilsea yendoana*) and four commercially available unsaturated fatty acids, 5,8,11,14-eicosatetraenoic acid (arachidic acid), 8,11,14-eicosatrienoic acid (dihomo-γ-linolenic acid), 6,9,12,15-octadecatetraenoic acid and γ-linolenic acid **15** on *Monostroma oxyspermum*. The MICs of the polar acids were all 1 µg/mL, except 8,11,14-eicosatrienoic acid (10 µg/mL) and palmitic acid 8 inactive. Alamsjah and coworkers [59] determined the toxic effect of α-linolenic acid and linoleic acid **14**, isolated from *Ulva fasciata* and *H. akashiwo*. The lethal concentrations (LC_50_) of these two acids were 0.58 and 1.91 µg/mL. They then suggested that α-linolenic acid had higher toxicity than linoleic acid 14 due to a feature of α-linolenic acid’s chemical structure, such as the number of unsaturated double bonds.

Nakai and coworkers [60] studied the anti-cyanobacterial activity of the aquatic plant *Myriophyllum spicatum* on *Microcystis aerugimosa*. The active fractions contained the polyphenols, gallic acid, pyrogallic acid, ellagic acid, and (+)-catechin, and the fatty acids, nonanoic acid, tetradecanoic acid, hexadecanoic acid, octadecanoic acid 11, 6-*cis*-octadecenoic acid, and 9-*cis*-octadecenoic acid. Pure nonanoic acid demonstrated the most potent inhibitory effect (EC_50_ value of 0.5 mg/L, and the EC50 values of the other pure compounds were 3.3 mg/L (6-*cis*-octadecenoic acid), 1.6 mg/L (9-*cis*-octadecenoic acid), 1 mg/L (gallic acid), 0.7 mg/L (pyrogallic acid), 5.1 mg/L (ellagic acid) and 5.5 mg/L ((+)-catechin), all significantly demonstrating inhibition of *M. aeruginosa* growth. In contrast, tetradecanoic acid, hexadecanoic acid, and octadecanoic acid did not show any effect at 10 mg/L. Nonanoic acid’s total inhibitory effect on *M. spicatum* culture was unexpected because the apparent concentration of nonanoic acid in the *M. spicatum* culture solution was 50 µg/L, much lower than the EC_50_. They suggested that because polyphenols and fatty acids have different chemical properties, resulting in further growth inhibition modes, it is reasonable to expect that their cyanobacterial growth inhibition activities may be additive or synergistic. In addition, the inhibitory effect of 9-*cis*-octadecenoic acid was more potent than 6-*cis*-octadecenoic. Among the four saturated fatty acids identified, nonanoic acid, with the shortest carbon chain, was the only one that demonstrated significant growth inhibition of *M. aeruginosa* at 0.5 mg/L.

The allelopathic effects of norisoprenoids, particularly dihydroactinidiolide 10, were first reported by Stevens and Merrill [61]; this compound isolated from the spike rush plant inhibited reddish seed germination. In 1993, Kato-Noguchi and coworkers [40] separated *R*-(−)-3-hydroxy-β-ionone from *Phaseolus vulgaris* and found that this compound inhibited the hypocotyl growth of lettuce seedlings and hypocotyls segments of dwarf bean seedlings at concentrations greater than 0.3 µM. Macías and coworkers [45] isolated allelochemicals from sunflower leaves and observed that 3-hydroxy-5β,6β-epoxy-β-ionone 19 demonstrated the greatest effects over monocotyledon species (*Hordeum vulgare* and *Allium cepa*). Moreover, Macías and coworkers [62] examined the allelopathic effects of twelve phenolics, two loliolides (isololiolide and loliolide 20), a diterpene, and a cyclitol isolated from the MeOH extract of the herb *Melilotus messanensis* on tomato (*Lycopersicum esculentum*) and barley (*Hordeum vugare*). They found that both loliolide 20 and isololiolide inhibited the germination of tomatoes and promoted the germination of barley. D’Abrosca and coworkers [42] isolated twelve C_13_ norisoprenoids from the leaves of *Cestrum parqui* (Solanaceae). All compounds were tested for allelopathic activity on seed germination and *Lactuca sativa* L seedling growth. Except for the C_13_ norisoprenoid, the compounds did not affect germination. However, they moderately inhibited lettuce root and shoot growth. Xian and coworkers [63] studied the allelopathic potential of submerged macrophyte *Vallisneria spiralis* Linn on *M. aeruginosa* Kütz. Analysis of the two active subfractions (extracted with chloroform), by high-resolution GC-MS, revealed that one subfraction consisted of 2-ethyl-3-methylmaleimide (77.10%), dihydroactinidiolide 10 (18.90%), and 4-oxo-β-ionone 13 (4.00%). The second subfraction consisted of 3-hydroxy-5,6-epoxy-β-ionone (38.30%), loliolide 20 (10.10%), 6-hydroxy-3-oxo-α-ionone (44.50%), and unknown (7.10%, MW 289). 

It has been reported that subfractions and isolated pure compounds exhibited significant inhibitory effects on the tested plant seeds and roots more than shoots because roots were in direct contact with inhibitors [6]. At a higher concentration, the extracts and pure compounds showed an inhibitory effect, but at a lower concentration, growth was stimulated. These results are consistent with other studies that showed that some allelochemicals demonstrate a stimulatory effect at low concentrations [64,65]. A previous comparison of the impact of allelopathic compounds on barnyardgrass and Chinese amaranth revealed that the extracts and pure compounds had a greater inhibitory effect on Chinese amaranth than barnyardgrass. It was reported previously that the mass and size of the seeds tested affect allelopathic activity [66] and that larger seeds are more tolerant than smaller seeds. The effects of allelochemical compounds from *S. platensis* against the tested plants in this study were consistent with that observation. Notably, barnyardgrass demonstrated a strong resistance against phytotoxins released by plants other than paddy weeds [67]. In the present study, the allelopathic activity of subfractions from E6, commercial fatty acids, isolated dihydroactinidiolide **10**, and 4-oxo-β-ionone **13**, dihydroactinidiolide **10** showed the highest activity. These compounds exhibited activity similar to subfraction E6F4, but the primary fatty acids such as linoleic acid **14**, showed relatively lower activity. This may be due to the synergy of all compound mixtures of fatty acids, fatty acids and 2-ethyl-3-methylmalaimide **9** fatty acids and 4-oxo-β-ionone **13** or fatty acids and dihydroactinidiolide **10** because the synergy between aromatic compounds and dihydroactinidiolide 10 or aromatic compounds and fatty acids have already been reported [60,68,69].

## 4. Materials and Methods

### 4.1. General Experiment Procedures

All solvents used in this study were available from Italmar Co., Ltd. (Bangkok, Thailand). All commercial fatty acids were purchased from Sigma-Aldrich (Singapore). The structure and purity of all identified compounds were confirmed by FT-NMR, and GC-MS analyses. ^1^H and ^13^C NMR spectra were provided by a Bruker AVANCE 300 NMR spectrometer (Bruker BioSpin AG, Fällanden, Switzerland), operating at 300 MHz and 75.5 MHz, respectively. FCC and quick column chromatography (QCC) were conducted by using Merck silica gel 60 (<0.063 mm) and Scharlau GE 0048 silica gel 60 (0.02–0.06 mm), respectively. For TLC, MERCK precoated silica gel 60 F_254_ plates were used. Spots on TLC were visualized under UV light and sprayed with an anisaldehyde-sulfuric acid reagent, then heated.

### 4.2. Algae Material and Preparation of Crude Extracts

Dried *S. platensis* was obtained from the Department of Fisheries, School of Agricultural Technology, King Mongkut’s Institute of Technology Ladkrabang. Algal cultures were grown in Zarrouk medium [70] and maintained in a 0.03% CO_2_ atmosphere at 25 °C and pH 10.5. The cultivated flasks were illuminated under 400 μmol m^−2^ s^−1^ light intensity. Cells were harvested at the late exponential phase by centrifugation and dried in an oven at 40 °C. The dried *S. platensis* (4.0 kg) were extracted successively by using *n*-hexane treatment for seven days at room temperature. The extract was then filtered through a Whatman No. 1 paper. The collected filtrate was evaporated to dryness under reduced pressure at 40 °C by using a rotary evaporator to yield crude *n*-hexane extract (25.9 g). The residue was then similarly extracted with ethyl acetate (EtOAc) or methanol (MeOH) to produce crude ethyl acetate (125.9 g) and methanol (795.6 g) extracts, respectively.

### 4.3. The Preparation of Fatty Acid Methyl Esters

Subfractions, E6F3-E6F5, (30 mg) were methylated with 2% H_2_SO_4_ in MeOH (20 mL), and the resulting solution was stirred at room temperature for 30 min. The mixture was heated until reflux at 80 °C for 2 h, and the reaction was completed by TLC. The mixture was cooled to room temperature, and excess methanol was removed under reduced pressure. The crude product was dissolved with ethyl acetate (20 mL) and washed with 0.5 M NaHCO_3_ (2 × 40 mL), then brine (10 mL). The product was dried (Na_2_SO_4_), filtered through Whatman No. 1 filter paper, and evaporated to dryness under reduced pressure to yield the fatty acid methyl ester products. These products were prepared to a concentration of 1500 ppm in ethyl acetate before GC-MS analysis. 

### 4.4. GC-MS Conditions

The GC-MS analyses were performed on an Agilent 6890 GC (Wilmington, DE, USA) equipped with a flame ionization detector. The automated split injection was executed by using an Agilent 7683 autosampler (Santa Clara, CA, USA). The total run time of the active fractions was 30 min. The GC oven temperature program was as follows: 50 °C hold for 1 min, 25 °C/min increase to 230 °C, and 13 min hold. Fatty acid methyl esters (FAMEs) and other compounds were identified by comparing the retention time of the integrated peak of a specific FAME with those of authentic standard FAMEs and the databank of the Wiley 7N Edition (Agilent Part No. G1035B). The height of less than 1% is not qualified, and the mass range was scanned from 50 to 500 a.m.u.

### 4.5. Tested Plants

Chinese amaranth (*Amaranthus tricolor* L.) seeds were purchased from the Thai Seed & Agriculture Co., Ltd., (Bangkok, Thailand) and barnyardgrass (*Echinochloa crus-galli* (L.) Beauv.) seeds were randomly collected from rice fields in Phitsanulok province, Thailand in August 2019. The germination rate of the test seeds was >80%.

### 4.6. Allelopathic Assay 

All subfractions were dissolved in ethyl acetate to compare their allelopathic effects. A total of 50 μM microliters of each fraction at concentrations of 10,000, 5000, 2500, and 1250 ppm were added to a glass vial (4.5 cm height × 2 cm diameter) lined with germination paper and evaporated to dryness for 3 h at room temperature. After evaporation, 0.5 mL of distilled water was added to the germination paper to yield the final concentrations at 1000, 500, 250, and 125 ppm. Then ten seeds of tested plants were placed on the germination paper as per treatments. The control received only distilled water. The treatments were replicated four times in a completely randomized design. All vials were sealed with Parafilm® and maintained at 28–32 °C. After seven days, germination (%), shoot, and root growth were recorded for all treatments. Inhibition (%) relative to control was calculated as follows [18]:Inhibition (% of control) = {100 − [*Sample extracts/Control*) × 100]}(1)

## 5. Conclusions

The crude ethyl acetate extract from the cyanobacteria *S. platensis* showed the highest inhibitory activity in this study. Purification and identification of the crude extract revealed that the major active compound was linoleic acid **14**. In addition to fatty acids, the extract also contained nonfatty acids which can play an allelochemical role, including 2-ethyl-3-methylmaleimide **9** and six norisoprenoids: dihydroactionidiolide **10**, 4-oxo-β-ionone **13**, 3-hydroxy-β-ionone **17**, 3-hydroxy-5α,6α-epoxy-β-ionone **18**, 3-hydroxy-5β,6β-epoxy-β-ionone **19**, and loliolide **20** all potent at low concentrations. Isolation of these nonfatty acid compounds from *S. platensis* is reported here for the first time. Our data confirm that *S. platensis* has allelopathic potential against the dicotyledon, Chinese amaranth, more than the monocotyledon, barnyardgrass. The results indicated that the inhibitory activity depended on the specific extract fraction and applied concentrations. The most robust compound was dihydroactinidiolide **10**. Allelochemicals from *S. platensis* could be developed into novel bioactive herbicides.

## Figures and Tables

**Figure 1 molecules-27-03852-f001:**
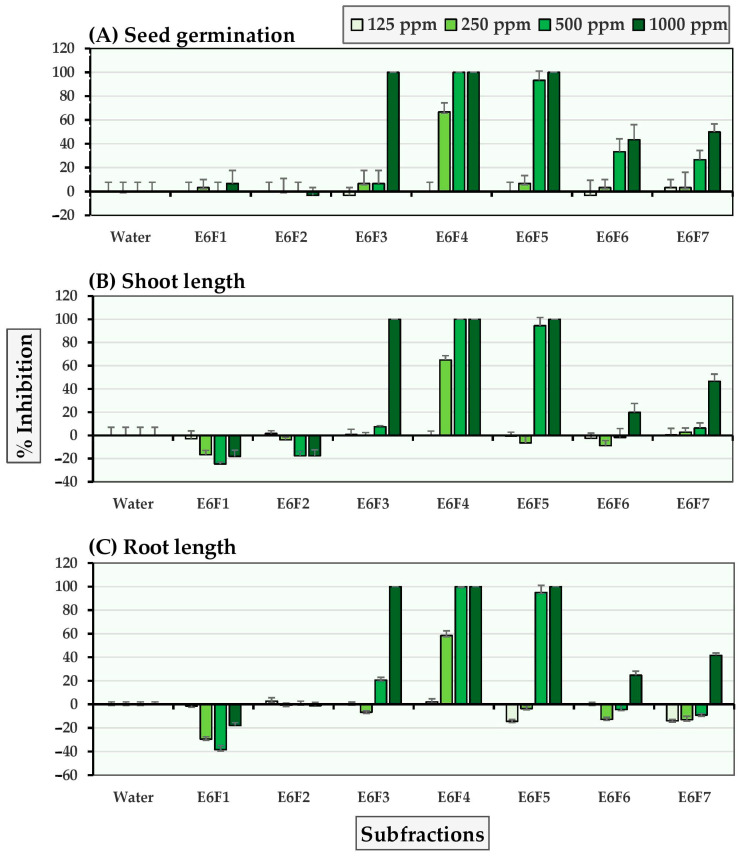
Inhibitory effects of *S. platensis* E6 subfractions on (**A**) seed germination, (**B**) shoot length, and (**C**) root length of Chinese amaranth. Distilled water was used as a control.

**Figure 2 molecules-27-03852-f002:**
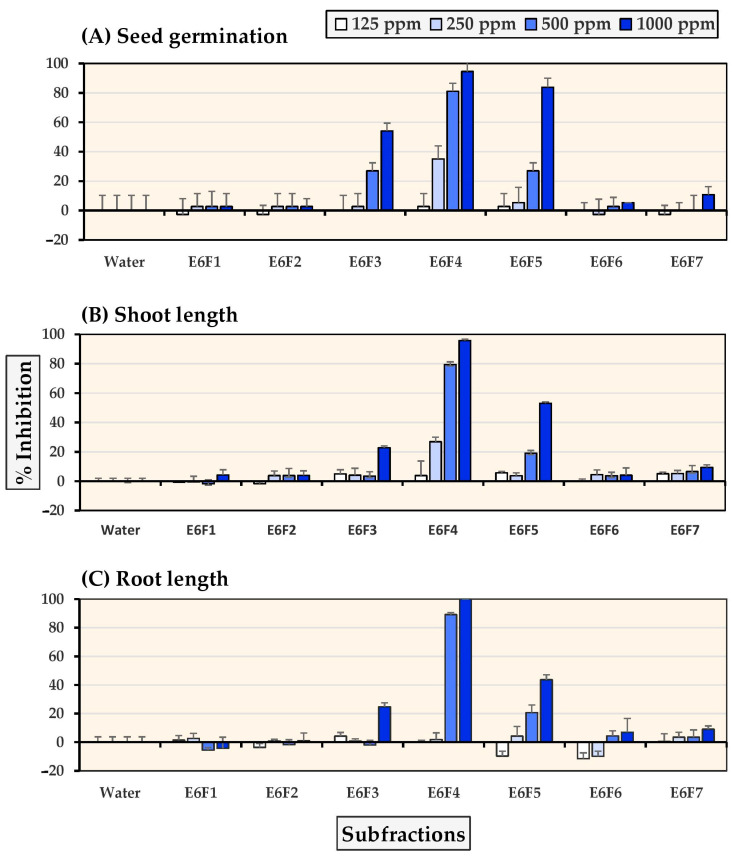
Inhibitory effects of *S. platensis* E6 subfractions on (**A**) seed germination, (**B**) shoot length, and (**C**) root length of barnyardgrass. Distilled water was used as a control.

**Figure 3 molecules-27-03852-f003:**
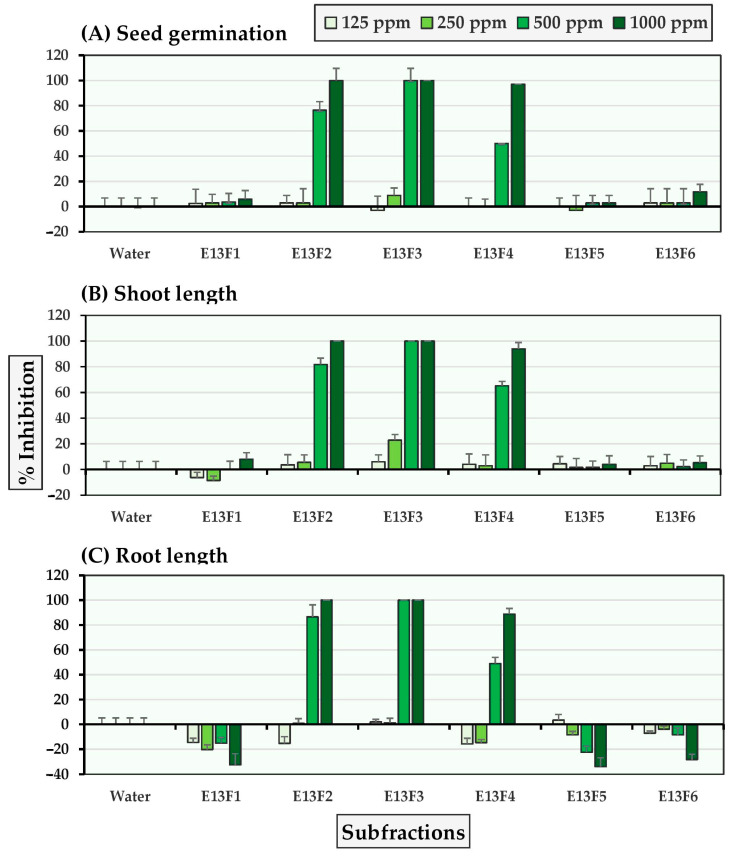
Inhibitory effects of *S. platensis* E13 subfractions on (**A**) seed germination, (**B**) shoot length, and (**C**) root length of Chinese amaranth. Distilled water was used as a control.

**Figure 4 molecules-27-03852-f004:**
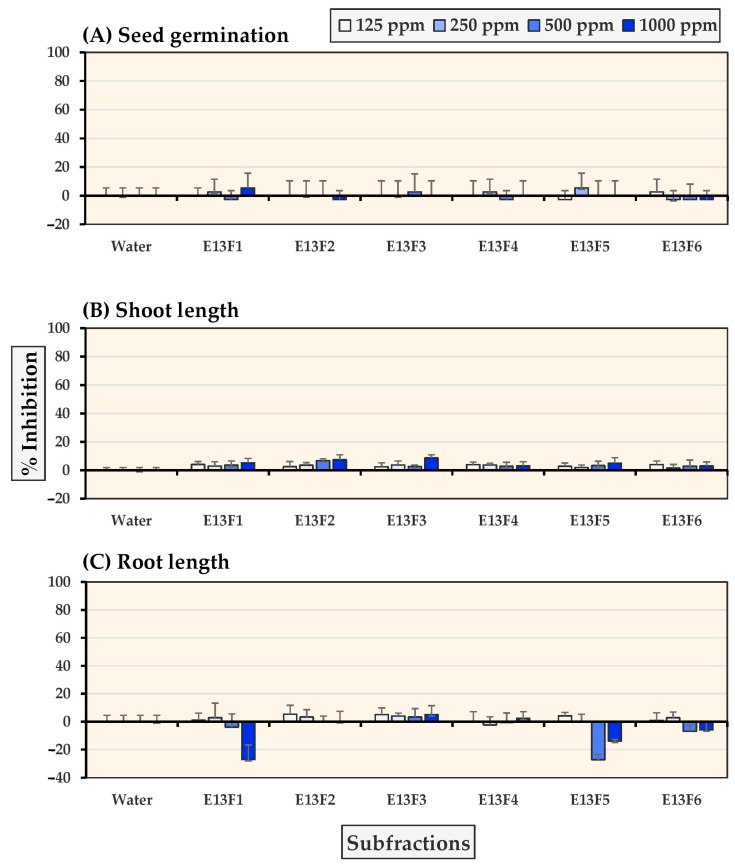
Inhibitory effects of *S. platensis* E13 subfractions on (**A**) seed germination, (**B**) shoot length, and (**C**) root length of barnyardgrass. Distilled water was used as a control.

**Figure 5 molecules-27-03852-f005:**
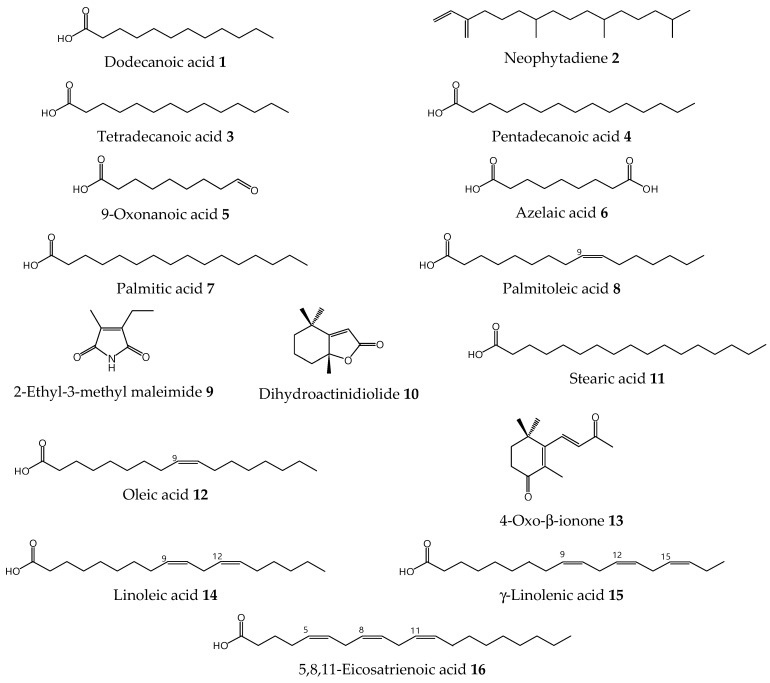
Compounds detected in allelopathic subfractions E6F3–E6F5 of *S. platensis*.

**Figure 6 molecules-27-03852-f006:**
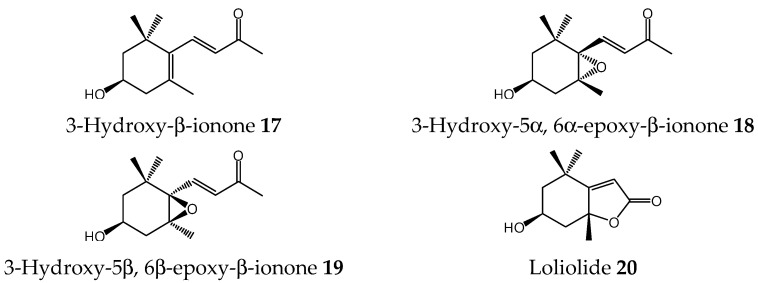
Compounds isolated from allelopathic subfractions E13F2-E13F4 of *S. platensis*.

**Figure 7 molecules-27-03852-f007:**
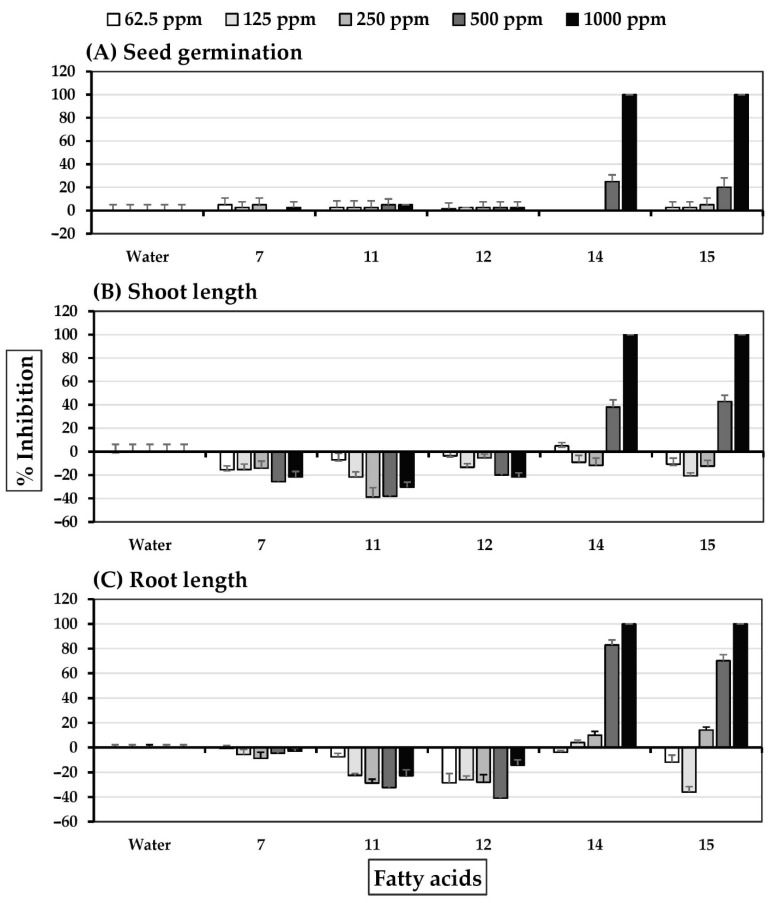
Inhibitory effects of fatty acids of *S. platensis* on (**A**) seed germination, (**B**) shoot length, and (**C**) root length of Chinses amaranth. Distilled water was used as a control.

**Figure 8 molecules-27-03852-f008:**
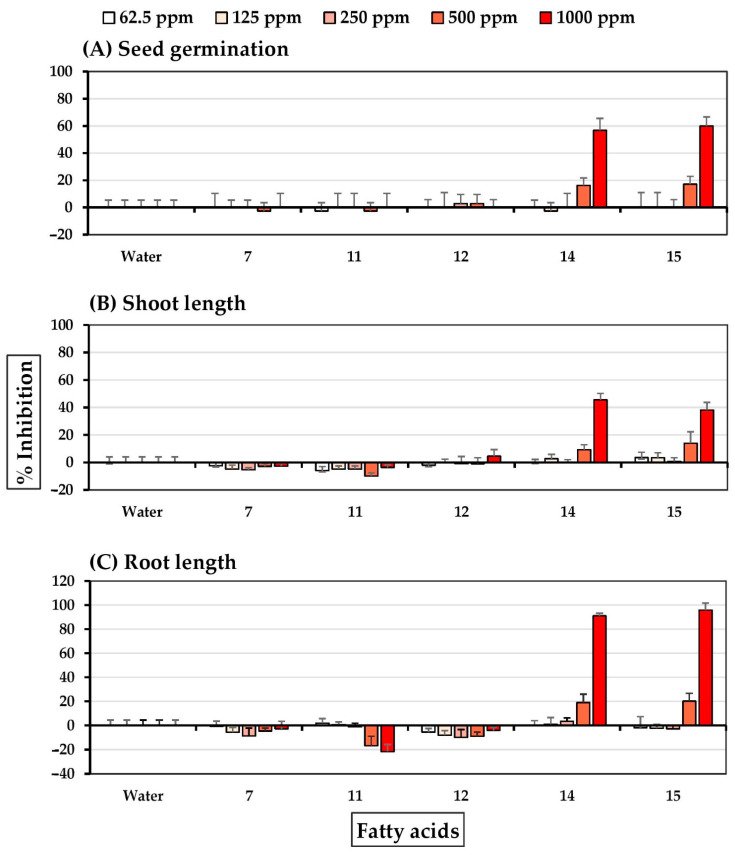
Inhibitory effects of fatty acids of *S. platensis* on (**A**) seed germination, (**B**) shoot length, and (**C**) root length of barnyardgrass. Distilled water was used as a control.

**Figure 9 molecules-27-03852-f009:**
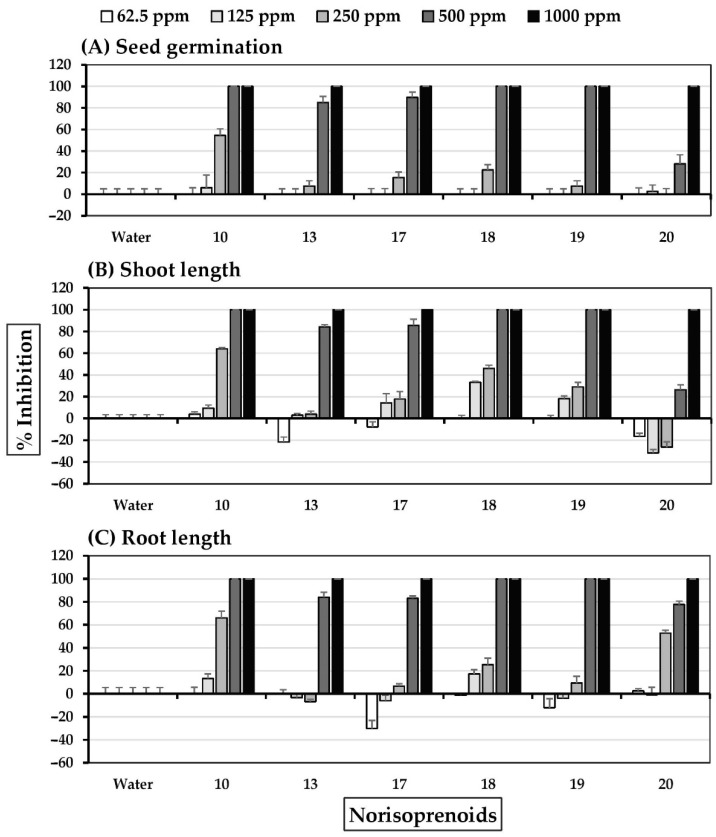
Inhibitory effects of norisoprenoids of *S. platensis* on (**A**) seed germination, (**B**) shoot length, and (**C**) root length of Chinese amaranth. Distilled water was used as a control.

**Figure 10 molecules-27-03852-f010:**
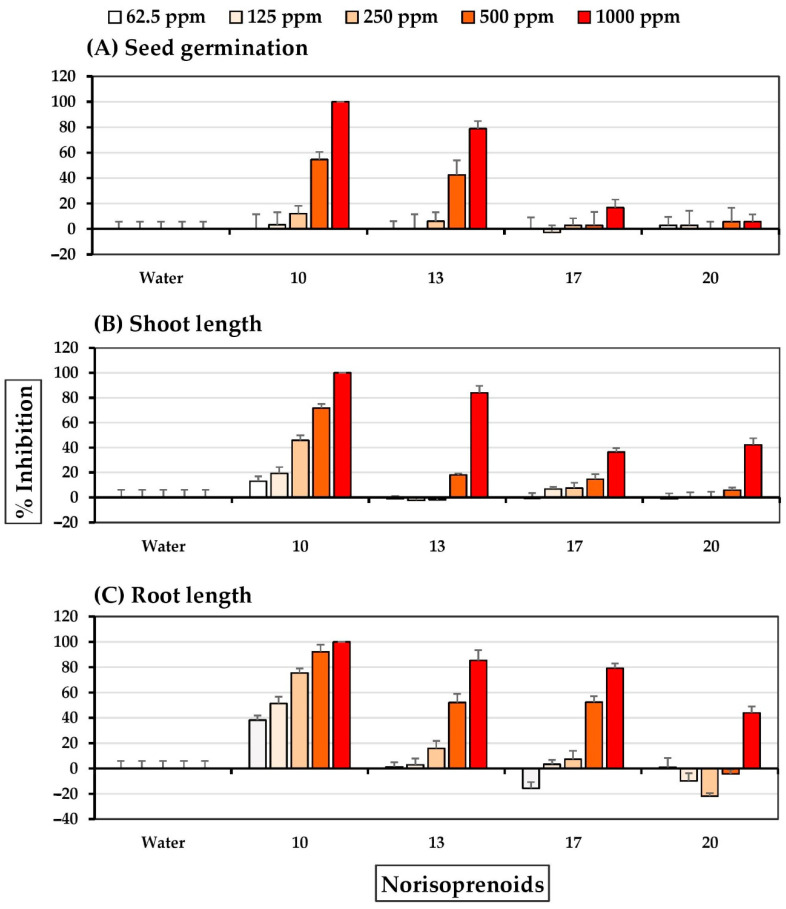
Inhibitory effects of norisoprenoids of *S. platensis* on (**A**) seed germination, (**B**) shoot length, and (**C**) root length of barnyardgrass. Distilled water was used as a control.

**Table 1 molecules-27-03852-t001:** Chemical compositions of active subfractions E6F3-E6F5 of *S. platensis*.

Compounds	GC Peak Area % ^a^		
	E6F3	E6F4	E6F5
Dodecanoic acid **1**	0.55	-	-
Neophytadiene **2**	-	-	1.04
Tetradecanoic acid **3**	3.97	-	0.41
Pentadecanoic acid **4**	2.16	-	-
9-Oxonanoic acid **5**	2.04	0.46	-
Azelaic acid **6**	1.60	-	-
Palmitic acid **7**	58.71	2.29	31.41
Palmitoleic acid **8**	12.82	4.55	2.71
2-Ethyl-3-methylmelaimide **9**	-	2.90	-
Dihydroactinidiolide **10**	1.27	8.32	-
Stearic acid **11**	1.69	0.42	1.87
Oleic acid **12**	11.76	1.17	3.16
4-Oxo-β-ionone **13**	1.66	2.15	1.91
Linoleic acid **14**	1.76	72.19	31.24
γ-Linolenic acid **15**	-	3.02	23.46
5,8,11-Eicosatrienoic acid **16**	-	2.53	2.80

^a^ Solvent peak is excluded.

## Data Availability

Not applicable.

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
