# Peer review of "The Allelopathic Activity of Extracts and Isolated from Spirulina platensis"

_molecules, 2022, doi:10.3390/molecules27123852_

Round 1

Reviewer 1 Report

In this study the allelopathic effects of crude organic (hexane, ethyl acetate, and methanol) extracts of the cyanobacterial, Spirulina platensis on barnyardgrass (Echinochloa crus-galli (L.) Beauv.) and Chinese amaranth (Amaranthus tricolor L.) were studied. The results is very interested but before publication a major revision is needed as reported in attached file.

Reviewer 2 Report

This manuscript described allelopathic activity-guided isolation and identification of the chemical constituents of Spirulina platensis, then evaluate the bioactivities of the isolated compounds. In my opinion, the work presented is interesting , however, I should suggest a few things to be consider by the author before manuscript is fully accepted.

What is positive control used in the allelopathic bioassay? As shown in this manuscript, the selection of the fraction and the subsequent isolation based on the result of the bioassay, therefore it is better to provide the corresponding data. Please check it throughout the whole manuscript.

Please keep one decimal for the coupling constants of the isolated compounds, and one writing style. (Page 7 Line 190 ‘dd, J = 17.61, 9.40 Hz’, Page 7 Line 205 ‘dd, J = 14.68 and 6.48 Hz’). And add one sentence ‘By comparison of their spectral data with those reported in the literature, this compound identified as …(compound name)’ following the list of NMR data for compounds 10, 13, 1720.

Aside, some minor corrections:

1. Page 7 Line 210:  ‘crytalyzed’    ‘crystallized’

2. Page 9 Line 263:  ‘determined four compounds’    ‘determined for compounds’

Reviewer 3 Report

The manuscript entitled “The Allelopathic Activity of Extracts and Isolated from Spirulina platensis”, authored by Patchanee Charoenying, Chamroon Laosinwattana, and Nawasit Chotsaeng, deals with the investigation of the allelopathic effects of crude organic (hexane, ethyl acetate, and methanol) extracts of the cyanobacterial, Spirulina platensis on barnyardgrass (Echinochloa crusgalli (L.) Beauv.) and Chinese amaranth (Amaranthus tricolor L.). The manuscript contains interested data that may find strong utility in screening particular compounds in this typology of extract. In addition, the revised manuscript is well written and organized. Perhaps the only problem is related to data presentation (poorly understood black and white histograms), which should be revised. In any case, I do not consider this to be a big problem for the suitability of the manuscript.

The following are a number of considerations that authors should fix before they can consider the manuscript suitable as a publication in Molecules.

1.       The abstract should be rewritten. In particular, as clearly explained in the authors' guidelines, a single paragraph of maximum 200 words is allowed. This section, should be written very briefly describing the current state of the art, the purpose of the work, the materials and method used, and then reporting the main data obtained. A concluding phase is also necessary.

2.       regarding the keywords, they are a useful tool to help indexers and search engines to find relevant papers of interest. If scientific search engines (such as PubMed, Scopus, Google Scholar, etc) can find a potential manuscript by the use of words contained in both title, abstract, and keywords. Consequently, readers will be able to find it too thank this words. An easier search of the manuscript allows to increase the number of people reading your manuscript after publication and, then, to obtain more citations. Consequently, keywords should be words preferably not contained in the title or abstract. This short explanation is to suggest that authors introduce as many keywords as they can, and replace those words that are already present at least in the title with new keywords properly related to the reviewed manuscript.

3.       Authors should include additional information in the introduction regarding the raw material used (Spirulina) during their experimentation. In particular, Spirulina is a fairly well-known raw material in the scientific and application worlds, as, due to its particular chemical composition, it is frequently used both as an ingredient in food supplements for human use (10.1016/j.jff.2019.103508) and within biostimulant formulations for agricultural use (10.1007/s10811-017-1242-z).

4.       Figure 1 and 2 should be made more understandable. For example,

a.       (i) the authors could divide the data into three different figures (one for each monitored parameter) to make the histograms more readable;

b.       (ii) make the graph more attractive by using colors, eliminating the gray lines in the background; MDPI does not cover additional costs for publishing colored images. So I strongly advise authors to use colors for their graphics.

c.       (iii) the authors should also remember to better complete the captioning of the figures, explaining exactly what is shown in the graph (mean and standard deviations?).

d.       (iv) Also, why was ANOVA not performed on these data? lowercase letters should be added in this graph according to the different significance of data.

5.       As mentioned earlier, spirulina is currently used as an ingredient in biostimulant formulations, which basically have an effect contrary to that described by the authors. The authors should in the discussion section extol and emphasize this concept, perhaps suggesting the disincentive of using this alga.

6.       It is unclear whether the paragraph spelled out between 176 - 216 is experimentally original to this article. If yes,

a.       (i) It should be reported in a separate and opportune annotated section;

b.       (ii) a materials and methods section related to the description of this methodology should be reproduced.

7.       Reading the materials and methods part, the authors exclusively restricted the metabolite research to fatty acids, having performed a transesterification (by the way, little described in the relevant section 4.3.). This particular technique, crucially causes the degradation of most molecules, allowing fatty acids to be visible by simple GC analysis. So my question is the following: did the authors perform the separation and identification of the different compounds described in result section before or after transesterification process? because if this was done after transesterification, the compounds they isolated and identified are adducts of reactions that might be absent in the original matter. Please, clarify this point.

Reviewer 4 Report

I really liked the manuscript!

 I would like to make a few remarks:

Please complete in the abstract with the final purpose of this study, the use of this important allelopathic action on invasive plants in crops.

Please specify why the two plants were chosen: barnyardgrass - Echinochloa crus-galli and Chinese amaranth - Amaranthus tricolor?

Round 2

Reviewer 1 Report

Accepted.